# Spending Time in the Forest or the Field: Investigations on Stress Perception and Psychological Well-Being—A Randomized Cross-Over Trial in Highly Sensitive Persons

**DOI:** 10.3390/ijerph192215322

**Published:** 2022-11-19

**Authors:** Katja Oomen-Welke, Evelyn Schlachter, Tina Hilbich, Johannes Naumann, Alexander Müller, Thilo Hinterberger, Roman Huber

**Affiliations:** 1Center for Complementary Medicine, Medical Center—University of Freiburg, Faculty of Medicine, University of Freiburg, Hugstetterstr. 55, 79106 Freiburg im Breisgau, Germany; 2Interdisciplinary Center for Treatment and Research in Balneology, European Institute for Physical Therapy and Balneology (EIPB), Sonnenbergstr. 35, 79117 Freiburg im Breisgau, Germany; 3Forschungsbereich Angewandte Bewusstseinswissenschaften, Abteilung für Psychosomatische Medizin, Universitätsklinikum Regensburg, Franz-Josef-Strauß-Allee 11, 93053 Regensburg, Germany

**Keywords:** forest environment, therapeutic landscape, anxiety, depression, stress, highly sensitive persons, relaxation, forest bathing, Shinrin-Yoku

## Abstract

Research suggests that stays in a forest promote relaxation and reduce stress compared to spending time in a city. The aim of this study was to compare stays in a forest with another natural environment, a cultivated field. Healthy, highly sensitive persons (HSP, SV12 score > 18) aged between 18 and 70 years spent one hour in the forest and in the field at intervals of one week. The primary outcome was measured using the Change in Subjective Self-Perception (CSP-14) questionnaire. Secondary outcomes were measured using the Profile Of Mood States (POMS) questionnaire and by analyzing salivary cortisol. We randomized 43 participants. Thirty-nine were allocated and included in the intention-to-treat analysis (90% female, mean age 45 years). CSP-14 in part showed significant differences—total score (*p* = 0.054, Cohen’s d = 0.319), item “integration” (*p* = 0.028, Cohen’s d = 0.365)—favoring the effects of the forest. These effects were more pronounced in summer (August). In October, during rainfall, we detected no relevant differences. POMS only showed a significant difference in the subcategory “depression/anxiety” in favor of the field. The amount of cortisol in saliva was not different between the groups. A short-term stay in a forest in summer caused a greater improvement in mood and well-being in HSP than in a field. This effect was not detectable during bad weather in the fall.

## 1. Introduction

The medicinal use of forests is of increasing interest worldwide. Forest air is refreshing because trees clean the air of pollutants such as nitrogen oxides and sulfur oxides, produce oxygen, and release volatile bioactive terpenes into the air [1]. Research from Japan [1], South Korea [2], China [3], Taiwan [4], Australia [5], the United States [6], Italy [7], Denmark [8], Norway [9], the United Kingdom [10], Luxembourg [11], Iceland [11], Finland [12], Sweden [13], Hungary [14], Germany [9] and Austria [15] suggests that spending time in the forest promotes relaxation, lowers stress hormones and blood pressure [16] and strengthens the immune system [17]. Most studies compared stays in the forest to stays in the city. Accordingly, forests potentially contribute to the prevention of stress-related diseases [18,19,20]. Controlled studies have shown positive effects in high blood pressure [21], chronic heart failure [22], COPD [23] and chronic neck pain [24]. In addition, spending time in the forest seems to improve psychological well-being [1].

Spending time in forests reduced adrenaline and noradrenaline in urine, cortisol in saliva and self-rated stress perception; it also induced relaxation in controlled trials [1]. This indicates that forest stays can reduce stress.

The available data also indicate that “forest bathing”, i.e., walking, standing or sitting in a forest with the purpose of relaxation, perceiving the environment and inhaling phytoncides stabilizes the autonomic nervous system by reducing the sympathetic and activating the parasympathetic tones [1]. With regard to the immune system, which is linked to stress response and vegetative nerve system, an increase in the activity of natural killer cells and the expression of anti-cancer proteins such as perforin, granzyme A/B, granulysin could be demonstrated [1]. In view of these findings, forests could make an important contribution to the prevention of stress-related diseases [19].

Many subjects reported a reduction in tension, anxiety, anger and fatigue as well as an increase in vigor after spending time in the forest [25]. Spending time in the forest relaxes and restores through sensory perceptions such as sounds, smells and visual impressions [1]. Environmental psychology has developed several theories for why nature has a positive impact on the human psyche, cognition, and attentional capacities. One of them is the reduction in stress [26] related to, among others, feeling safe in certain places (e.g., along watersides and when the horizon is visible) [27]. Another one is the restoration of attention [28]. Here, it is assumed that voluntary attention, e.g., in a city, requires cognitive control, which may become fatigued after prolonged use. The capacity for voluntary attention is replenished by exposure to nature [27]. Further theories combine the two abovementioned theories [16] or comprise individual aesthetic feelings, individual values and attitudes [29].

As shown in previous studies, the stressful environment of a city was most often compared to a forest; it remains unclear whether forests have specific effects or are just acting as natural environments. Therefore, we wanted to compare two natural but polar-opposite environments. In cultivated fields, sensory impressions are different from the forest. In order to maximize profitability [30], fields are mostly structured into rectangular shapes and usually mainly one type of plant is found, while in a natural forest, different types of plants grow side by side [31]. Accordingly, visual, auditory and olfactory impressions are less diverse in fields than in forests.

The play of light and shadow that characterizes the forest atmosphere is not found in fields. The plants are usually not tall enough to provide shade, whereas the height of the trees in the forest can provide a sense of shelter.

Field paths are more often sealed than forest paths, which changes haptic perception when walking on them. Thus, there are significant differences in the types of sensory impressions between forests and fields.

Highly sensitive persons (HSP), due to their subtle perception, intensely perceive stimuli that others might not even consciously notice. These stimuli may consist of the behavior or moods of other people, the media, medications, pain, and hunger [32]. They perceive stimuli, positively or negatively, to a higher degree, which may, on the one hand, lead to a prolonged reaction time [33], and on the other hand to more intense feelings and emotional excitability [32].

Due to their low threshold of perception, they often feel inundated by stimuli, suffer stress more quickly than other people, and are more easily aroused emotionally [34,35]. Strategies for avoiding negative feelings are shyness, introversion, and social withdrawal [33].

We expected the differences between the two natural environments to be subtle. Because of their low threshold for stimulation and strong reactions to low levels of stimuli, HSPs can be expected to respond more clearly to a changing environment. Therefore, it is likely that the differences in the effects of forest and fields are more pronounced for people with HSP than for the general population. In addition, people with HSP feel stressed more quickly, so the calming effect of the forest should be particularly effective for them.

The aim of the study was to investigate the effects of spending time in the forest and the field of psychological well-being, stress and anxiety in HSP. We suspected that people with above average sensitivity would react particularly intensively to stays in the forest and that these stays could have a positive effect on mood and well-being.

## 2. Materials and Methods

In this crossover, randomized, controlled trial, we included men and women 18–70 years of age with HSP, defined as a total sensitivity (sum of items 1–6) of >18 points measured using the Sensitivity and Processing Questionnaire (SV12) [35]. To differentiate high sensitivity from mental illness, we used the ICD10 Symptom Rating Scale (ISR) for screening and excluded subjects with high symptom load (ISR ≥1.7). Subjects with an ISR <0.5 were included without further investigation; subjects with an ISR ≥0.5 (suspected mental illness) and 1.6 (moderate symptom load) were evaluated by a physician experienced in psychotherapy and included if mental illness was not suspected. Self-reported serious concomitant physical or mental diseases, pregnancy, lactation period, lack of compliance, alcohol or drug abuse, participation in another intervention study in the past 4 weeks, inability to speak German and inability to walk 1–2 km were further exclusion criteria.

We registered the study in the German Clinical Trials Database (DRKS00020787) and conducted it in accordance with Good Clinical Practice Guidelines (CPMP/ICH/135/95; Topic E6 (R1); and GCP-V), the Declaration of Helsinki and local laws. The local ethics committee reviewed and approved the protocol (EK-Freiburg registration number 70/20). All participants provided written informed consent before participation.

After screening by telephone, potentially eligible subjects were assessed by a physician experienced in psychotherapy, provided written informed consent, were checked for inclusion and exclusion criteria again and, if eligible, randomized into groups of up to seven subjects. We chose a forest site and fields in comparable accessibility within 30 min by car or public transport from the University of Freiburg. We recommended sturdy shoes and clothing appropriate for the weather. On day 1, Group 1 met at the forest and Group 2 at the field, respectively. Immediately upon arrival, the subjects of both groups completed the POMS questionnaire. After a short walk, the subjects spent time walking, standing or sitting, perceiving the surroundings without being instructed to perform specific exercises. Each intervention lasted about one hour. Then, we collected saliva samples on site, distributed CSP-14 and POMS questionnaires, and conducted qualitative interviews with randomly selected subjects (separate publication).

On day 8, Group 1 travelled to the field and Group 2 travelled to the forest. At both locations, subjects underwent the same program as on day 1.

The CSP-14 [36] is a feedback questionnaire developed to record changes in physical, emotional, and mental well-being after interventions. The CSP-14 consists of 19 items, 14 with a seven-point Likert scale, and 5 dichotomous questions. The participants filled in the CSP-14 immediately after the respective intervention. The primary outcome measure was a comparison of the total score and all three sub-categories (“balance”, “integration”, and “vitality”) between the interventions.

We used the German short version of the POMS [37], which is the most frequently used questionnaire in studies to assess emotional well-being. It can be used for variable observation periods and comprises 35 items with a seven-point response scale. The items are grouped into four scales: Depression/Anxiety, Fatigue, Vigor, and Hostility [37]. Total Mood Disturbance (TMD) is calculated from the sum of the subscales Depression/Anxiety, Fatigue, and Hostility minus the score for Vigor.

We administered the POMS before and after the respective intervention.

Furthermore, we collected saliva samples to determine the level of cortisol. The determination of cortisol in saliva is a recognized marker for the measurement of short-term stress levels [38,39]. To prevent bias due to the circadian change in cortisol levels during the course of the day, we took samples at the same time of day (10 a.m., after the respective intervention). The test persons each filled a 1.5 mL tube with saliva. The tubes were collected, transported to the University Medical Center Freiburg, frozen at −20 °C and analyzed in one batch by the Laboratory Clotten in Freiburg after the study was finished.

There was no information on the variance/standard deviation of the CSP-14 in our setting. Therefore, we used the standardized effect size d for this pilot trial and assumed a small-to-medium difference, d = 0.5, between forest and field; furthermore, a power of 80%, an alpha 5% and two-sided testing in a paired t-test cross-over design were used. We further assumed that there was no carry-over effect and no interaction between subjects, groups and periods. This resulted in a sample size of 35 per group. To adjust for possible dropouts (15%), we aimed for a group size of 40 subjects.

We numbered the participants by sequence of inclusion. The availability of subjects resulted in variable block sizes of five to fourteen. An independent, blinded researcher generated the randomization lists using https://randomization.com/ (accessed on 07 August 2020, 21 August 2020, 02 October 2020, and 16 October 2020) the day before the start of each pair of parallel groups. The participants were informed via email about the meeting point the evening before the intervention. Due to the nature of the interventions, further blinding was not feasible.

Data were entered blinded into the database. De-blinding was conducted after all data were entered and the database was closed. The primary analysis was based on intention to treat. The per-protocol population consisted of participants who completed at least 75% of the intervention as per the protocol. Missing values were replaced in SPSS using the Transform, Replace Missing Values, Linear Trend at Point function, which calculates a regression on an index variable scaled from 1 to *n* for the existing series. The missing values were replaced with the value predicted by regression.

A statistical analysis of the primary outcome criterion (difference in CSP-14 sum scores before and after intervention) was performed using a two-sided t-test for paired values between groups in a cross-over design. The cross-over design assumes similar variance in both groups.

A significant and clinically relevant result is assumed at a *p*-value < 0.05 and when the effect size is d > 0.5 according to Cohen. For the secondary outcome parameters, the effect sizes according to Cohen were reported with confidence intervals and otherwise analyzed descriptively. SPSS^®^, Version 25, for Windows was used as the analysis program. UEs were recorded and described.

## 3. Results

### 3.1. Study Population

The subjects were recruited between July and October 2020 through public notices, newspaper advertisements, and Facebook posts. The planned timeframe for the study was summer to fall 2020. We conducted the study on a total of eight weekend days in August and October.

We screened 150 interested persons through telephone interviews. Most of the respondents could not participate because suitable dates for the intervention could not be found (*n* = 82) or because they did not meet the inclusion criteria (SV12 ≤ 18 *n* = 1, age not fitting *n* = 10). Four subjects were excluded because of chronic physical disease and three due were excluded because they had an ISR score ≥ 1.7.

We recruited and randomized 43 subjects classified as highly sensitive persons according to the SV12 questionnaire (see Figure 1—flow diagram).

In each group, two participants dropped out before the first intervention due to intercurrent illness or time mismanagement unrelated to the study. Thirty-nine subjects participated in at least one intervention and were included in the analysis according to the intention-to-treat principle. After a wash-out period of one week, 37 people received the second intervention, and two had intercurrent illnesses.

Participants were aged between 18 and 69 years (mean 44.66 years, SD = 15.67). The collective consisted of 35 women and 4 men. Most (62%) of the participants worked full or part time, eight were college students and one was a high school student. All groups achieved an SV12 total sensitivity score of approximately 21, which is well above the population average of 16–18 (maximum score 24). In regard to the ISR score, all group mean scores were below the value of 0.5, which widely excludes psychiatric disease. Group characteristics are displayed in Table 1.

### 3.2. Treatment Efficacy

All parameters were normally distributed according to the Kolmogorov–Smirnov test.

The total CSP-14 score showed a higher mean value in favor of the forest (1.50 ± 1.1 versus 1.05 ± 0.99) in the ITT analysis (*n =* 39). The difference between field and forest was almost significant (*p* = 0.054) and showed a small effect (Cohen’s d = 0.319).

The sub-category “integration” (summarizing a relaxed and security-mediating body perception) showed a significant difference in favor of the forest (*p* = 0.028, mean 1.67 ± 1.03 versus 1.18 ± 1.04). However, the effect (d = 0.365) was small.

The sub-category “balance” (calmness, even temper) showed a difference of 0.41 points between forest (mean = 1.58 ± 1.26) and field (mean = 1.17 ± 1.24), which was not significant (*p* = 0.107, d = 0.264).

Regarding the sub-category “vitality”, again, higher values were found for the forest (mean = 1.22 ± 1.29) compared to the field (mean = 0.84 ± 1.00), but the difference was not significant (*p* = 0.156, d = 0.232).

In summary, the total collective showed better CSP-14 values for the forest, but a significant difference was only detectable in the sub-category “integration”. Figure 2 displays these results.

In the groups who had their intervention in summer (August), the differences were more pronounced (mean CSP total score forest = 1.44 ± 0.93 versus field 0.88 ± 0.98; *p* = 0.024; d = 0.504). Again, the strongest differences were found in the sub-category “integration” in favor of the forest (forest mean = 1.59 ± 0.85, field mean = 0.97 ± 1.05; *p* = 0.014; d = 0.555) but also in the sub-category “vitality” (forest mean = 1.24 ± 1.05, field mean 0.7 ± 0.94; *p* = 0.037; d = 0.464). Figure 3 provides a detailed overview.

In October (autumn, rainy weather), the CSP-14 values of all categories were still higher for the forest interventions, but the differences were smaller (Figure 4).

#### 3.2.1. POMS

In the total collective, POMS ratings were significantly better after both interventions when compared to before the interventions: “Total Mood Disturbance (TMD)” (*p* < 0.001, forest Cohen’s d = −0.765; field d = −0.654) and “Fatigue” (*p* < 0.001, forest d = −1.086; field d = −0.864). In the forest interventions, we found significant changes between T1 (before intervention) and T2 (after intervention) for “Vigor” (*p* = 0.002) and “Hostility” (*p* < 0.001). In the field interventions, there were significant changes in the scale “Depression/Anxiety” (*p* = 0.001). Figure 5 displays these changes.

When comparing the two interventions, we found no significant differences in “Total Mood Disturbance (TMD)”, nor in the scales “Vigor”, “Hostility”, or “Fatigue”. In the scale “Depression/Anxiety”, there was a significant group difference (*p* = 0.02) with small effect size (Cohen’s d = 0.39) in favor of the field intervention, which could in part be attributed to different baseline values before intervention (field mean = 11.62, forest mean = 7.51). Table 2 summarizes the POMS data of the total collective.

During summer (*n =* 23, Table 3), the forest groups beat the field groups in all scales, though differences were not significant. The comparison between T1 (before intervention) and T2 (after intervention) yielded mostly significant results for both forest and field intervention.

In autumn (Table 4), the forest intervention only showed a significant effect between T1 and T2 in the scales “Total Mood Disturbance (TMD)” (*p* = 0.049, d = −0.556) and “Fatigue” (*p* = 0.002, d = −0.966). In contrast, the field intervention led to a significant improvement in all scales. “Depression/Anxiety” was significantly different between the two interventions in favor of the field (*p* = 0.015, d = 0.716).

#### 3.2.2. Cortisol in Saliva

Due to the circadian differences, we decided against collecting baseline values. We found no significant differences between field and forest interventions, neither in summer, nor in winter or in the total collective (Table 5).

### 3.3. Safety, Tolerability and Compliance

During the interventions, the participants reported no side effects. During the wash-out period, three participants reported stress factors that were not related to the interventions (death of a relative, moving house and a wasp sting).

Four participants dropped out before the start of the study before having participated in any intervention. The reasons provided were acute illness (*n =* 2), weather conditions (*n =* 1) or mismanagement of time (*n =* 1).

During the study period, the overall compliance of the participants was good. Thirty-nine subjects received their first intervention as planned. There were two further drop-outs due to acute illness which were not related to the study. The drop-out rate of 5.13% was below the estimated rate.

## 4. Discussions

Spending time in the forest has proven beneficial effects on stress perception and psychological well-being when compared to a city [1,40]. Few studies compared two natural environments, and our study is the first examining stays in a forest with stays on a field in highly sensitive persons. According to our main results, both natural environments have a positive impact on the psychological dimensions measured with CSP-14 and POMS questionnaires, but that they seem to modify these dimensions differently.

Our main outcome results show that, as soon as one hour after entering the forest, participants felt a sense of security, relaxation and inner connectedness. In summer, forest interventions had a better effect on vitality. Our study was the first to use the CSP-14 questionnaire, and the comparisons between field and forest interventions were also novel. The differences between our two interventions were smaller than the differences between the forest interventions and urban environments examined in other studies. In 2006, Park et al. performed a cross-over study comparing the effects of walks in forests with urban areas in 168 male students in their early twenties. In the Japanese Version of the POMS, which consists of 30 items, the forest interventions significantly lowered perceptions of depression, anxiety, hostility, fatigue, confusion and total mood disturbance, and greatly increased vigor [1]. In 2019, Song et al. conducted a similar RCT with 60 women in their early twenties [40]. They found significant differences in all POMS subscales and the State-Trait Anxiety Inventory (STAI) in favor of the forest [40]. This greater difference in effects is not surprising, as urban environments differ much more from forests than fields with regard to light, smells, and sounds.

The influence of the season was not a primary focus of our study, but did come to our attention when we conducted a post hoc comparison of summer and autumn groups, which were separated by a holiday period of 5 weeks during which no recruitment was possible.

In autumn, depression and anxiety rated with the POMS were less intense after staying on the field, compared to the stay in the forest. The initial POMS ratings for “depression/anxiety” were, however, higher in the field than in the forest group, so no clear conclusions can be drawn.

Little is known from studies about the influence of weather and seasons on the health-related effects of a natural environment, despite this being an expected relationship based on our own experiences. Terpene emissions in forests vary between seasons [41]. Li et al. found better effects of a rainy forest trip than a sunny stroll through an urban area [42]. Some authors categorize weather conditions as irrelevant [43], whereas others decided not to conduct interventions in the rain [1,44]. Some studies assessed the effect of weather conditions on mood without consideration of the environment. Keller et al. stated that good weather was only related to a higher mood in spring [45]. Klimstra et al. defined four types of people in respect to their preferences: summer lovers, summer haters, rain haters and the unaffected [46]. Future studies could take these insights into consideration.

In our groups, the weather conditions were similar because they took place in parallel in the same area (distance 6 km and both 320 m above sea level). Therefore, a systematic bias in our results due to weather conditions can be excluded. Occasional in-parallel measurements showed small differences in temperature and humidity at the two intervention sites (e.g., on 29th August 20 °C and 72% in the forest and 18,5 °C and 66% in the field, respectively). Systematic measurements were not performed on each intervention day, which is a limitation.

As this was a pilot study, the sample size was not sufficient to be confirmatory; therefore, the results require further substantiation. There was an imbalance in gender with only four male participants, even though the difference in the prevalence in men and women is not that high [47]. Women are known to participate more frequently in psychological studies than men [48]. Our results, therefore, are less representative of men than women. In other studies, men showed a substantial improvement in mood [49].

Blinding was not possible due to the nature of the interventions. Interventions took place in August and October, during two seasons of the year, which turned out to be a relevant confounder and provided an opportunity to study seasonal influences. Each intervention took place once for one hour. Longer stays and the sustainability of the effects were not investigated in this study. The site of the field intervention had a nice view over mountains and forests, which might have caused a contamination bias. The forest, on the other hand, was close to the city and had, therefore, some visitors, which impaired the peace and calm that might be found in forest areas further away. Last but not least, there is no generally accepted definition of high sensitivity. We tried to solve this issue pragmatically by using a validated questionnaire to discriminate between high and normal sensitivity.

Using a cross-over design compensated confounding factors such as age, gender and, in the case of the present study, the expression of high sensitivity. In order to compensate for the influences of the investigators, we assigned the same two guides to each group for both interventions. Conducting the interventions in parallel groups only a few kilometers apart allowed for the best possible control of influencing factors such as weather conditions. Using this design, we achieved the maximum plannable internal validity.

This study shows that forests are not the only kind of natural environment that can promote psychological well-being. The characteristics and qualities of natural environments might influence people’s mood and well-being differently. There might also be differences dependent on the preferences of the respective individuals. We regard it as meaningful to study these different effects of nature on the human soul and body in more detail. In addition, future studies examining the effects of different natural environments on human health should respect seasonal aspects and weather conditions.

## 5. Conclusions

Both stays in the forest and in the field result in improved emotional well-being measured with a POMS questionnaire. CSP-14 total scores and especially feelings of security and vitality were better after staying in the forest compared to staying on a field. The intensity of these effects is probably modified by the season and the weather.

## Figures and Tables

**Figure 1 ijerph-19-15322-f001:**
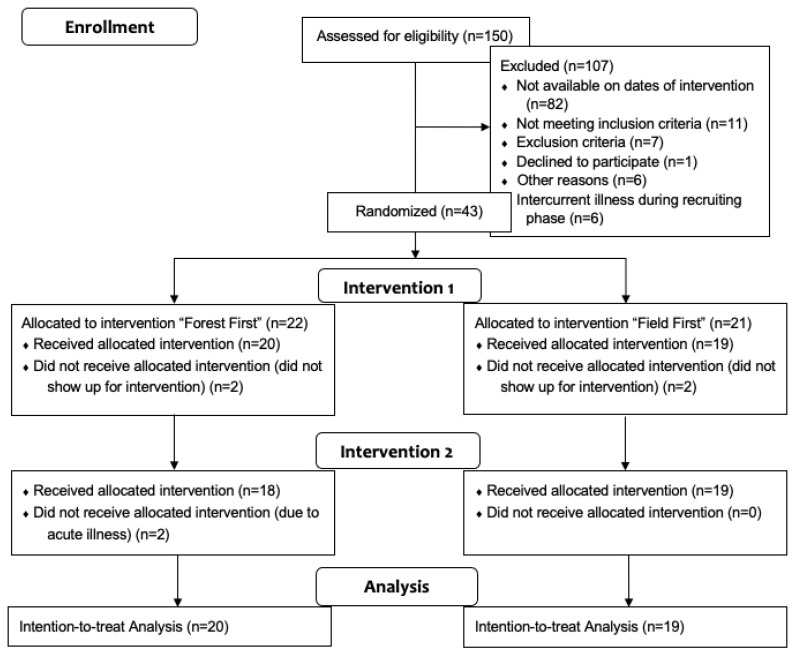
Flow diagram.

**Figure 2 ijerph-19-15322-f002:**
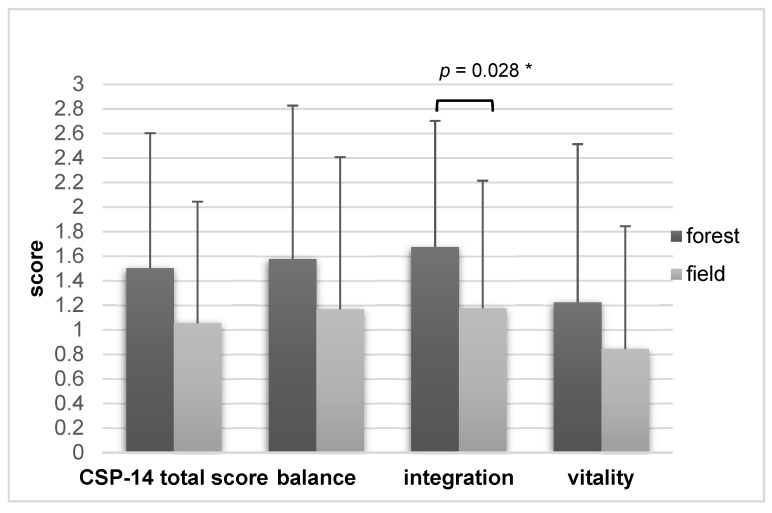
CSP-14 of the ITT total collective (*n =* 39, mean values ± standard deviation), comparison of forest and field interventions. * = *p* < 0.05.

**Figure 3 ijerph-19-15322-f003:**
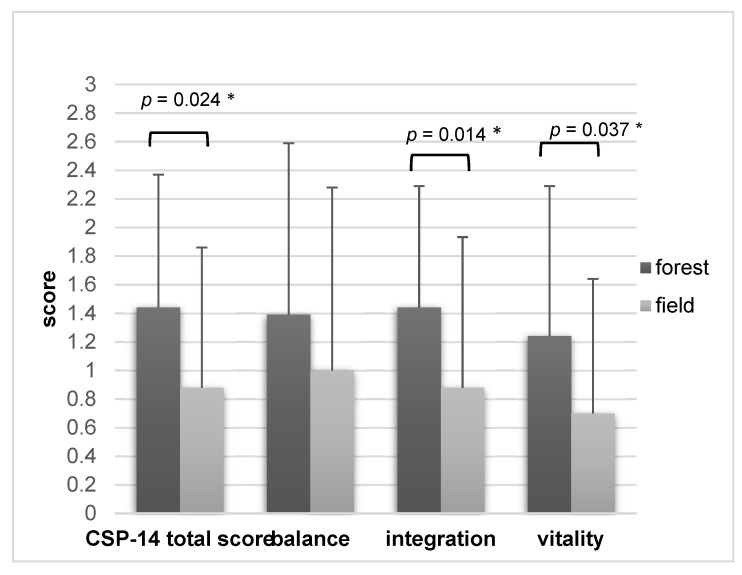
CSP-14 summer collective (*n =* 23, mean values ± standard deviation), comparison of forest and field interventions. * = *p* < 0.05.

**Figure 4 ijerph-19-15322-f004:**
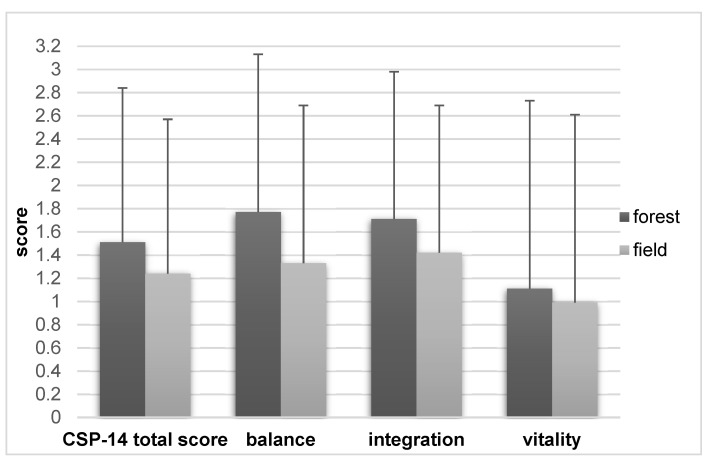
CSP-14 of the ITT autumn collective (*n =* 15, mean values ± standard deviation), comparison of forest and field interventions.

**Figure 5 ijerph-19-15322-f005:**
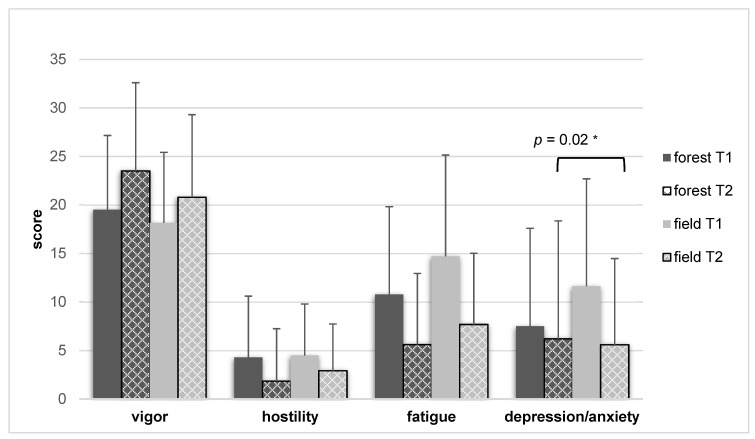
ITT analysis of the POMS (*n =* 39, mean values ± standard deviation), and comparison of forest and field interventions. Higher values show improvement in vigor; lower values show improvement in hostility, fatigue, depression/anxiety. T1 = before, T2 = after intervention, *p* = significance level of the difference T2-T1, * *p* < 0.05.

**Table 1 ijerph-19-15322-t001:** Group characteristics.

Group Characteristics	FoFi	FiFo	Total
Participants (n)	20	19	39
Sex (f/m)	19/3	20/1	35/4
Age (years)	45.6 ± 16.1	43.7 ± 15.6	44.7 ± 15.7
SV12 total sensitivity	21.1 ±1.3	20.9 ± 1.5	21.0 ± 1.4
ISR total score	0.29 ±0.26	0.44 ± 0.37	0.36 ± 0.32
ISR symptom load (none/small/medium/heavy)	17/2/1/0	13/5/1/0	30/2/5/0
Employed (e/s/ue/rt)	11/4/1/4	13/5/0/1	24/9/1/5

FoFi = first forest, then field, FiFo = first field, then forest, f = female, m = male, e = employed, s = student, ue = unemployed, rt = retired; SV12 16-18 average sensitivity, max. score 24; ISR ICD-10 symptom rating total score (<0.5 no psychiatric disease).

**Table 2 ijerph-19-15322-t002:** Profile of Mood States (POMS *n* = 39, mean ± SD). TMD = Total Mood Disturbance.

POMS	Forest T1	Forest T2	Field T1	Field T2	*p*-Value	Cohen’s d
TMD	3.05 ± 29.8	−9.74 ± 29.8	12.68 ± 28.74	−4.59 ± 23.96	0.462	0.119
Vigor	19.51 ± 7.65	23.51 ± 9.09	18.14 ± 7.28	20.78 ± 8.52	0.482	0.114
Hostility	4.28 ± 6.32	1.85 ± 5.40	4.49 ± 5.30	2.92 ± 4.82	0.35	0.152
Fatigue	10.77 ± 9.05	5.62 ± 7.31	14.7 ± 10.44	7.68 ± 7.33	0.185	0.216
Depression	7.51 ± 10.08	6.21 ± 12.15	11.62 ± 11.07	5.59 ± 8.88	0.02	0.39

**Table 3 ijerph-19-15322-t003:** Profile of Mood States (POMS) summer collective (*n =* 23, mean ± SD). TMD = Total Mood Disturbance.

POMS	Forest T1	Forest T2	Field T1	Field T2	*p*-Value	Cohen’s d
TMD	−0.96 ± 18.43	−14.13 ± 16.3	11.36 ± 27.85	0.91 ± 26.12	0.683	−0.086
Vigor	20.09 ± 6.63	23.52 ± 7.8	18.32 ± 7.3	18.64 ± 8.7	0.052	0.428
Hostility	3.13 ± 3.55	0.70 ± 1.18	4.59 ± 5.02	3.82 ± 5.77	0.242	0.251
Fatigue	9.78 ± 6.84	4.48 ± 4.57	14.09 ± 10.21	8.91 ± 7.67	0.934	0.017
Depression	6.08 ± 6.28	3.87 ± 6.43	11.0 ± 10.0	6.82 ± 9.59	0.403	0.178

**Table 4 ijerph-19-15322-t004:** Profile of Mood States (POMS) autumn collective (*n =* 15, mean ± SD). TMD = Total Mood Disturbance.

POMS	Forest T1	Forest T2	Field T1	Field T2	*p*-Value	Cohen’s d
TMD	10.07 ± 41.06	−1.67 ± 43.16	14.6 ± 30.88	−12.67 ± 18.31	0.27	0.296
Vigor	18.0 ± 8.96	23.0 ± 11.14	17.87 ± 7.5	23.93 ± 7.43	0.973	−0.009
Hostility	6.07 ± 9.11	3.73 ± 8.4	4.33 ± 5.86	1.6 ± 2.59	0.747	0.085
Fatigue	12.2 ± 12.02	7.67 ± 10.2	15.6 ± 11.07	5.87 ± 6.64	0.108	0.443
Depression	9.8 ± 14.21	9.93 ± 17.57	12.53 ± 12.79	3.8 ± 7.67	0.015	0.716

**Table 5 ijerph-19-15322-t005:** Cortisol (µg/dL) in saliva (*n =* 39).

Collective	Forest	Field	*p*	Cohen’s d
Total	0.13 ± 0.08	0.11 ± 0.06	0.114	0.259
Summer	0.13 ± 0.06	0.12 ± 0.07	0.725	0.074
Autumn	0.14 ± 0.09	0.09 ± 0.04	0.065	0.517

## Data Availability

Not applicable.

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
