# Peer review of "Spending Time in the Forest or the Field: Investigations on Stress Perception and Psychological Well-Being—A Randomized Cross-Over Trial in Highly Sensitive Persons"

_ijerph, 2022, doi:10.3390/ijerph192215322_

Round 1

Reviewer 1 Report

Title: Is there any possibility to shorten the title.  I found it very long and difficult to read.

Line 16: Another natural environment - could you maybe specify? Is it an open landscape? Water environment?

Line 49: Forets bathing - I think that this needs to be specified more as not all readers will understand that it goes beyond just having a walk in the forest.

Line 57-59: Here you can add some elements of environmental psychology. There are around 6 theories describing why nature has a positive effect, such as e.g., the fractals that they present. I Think in the context of stress diseases this information could be very good to use.

Line 60: Cultivated field - similar should be used in the abstract.

Line 65-69: could be referenced with some env. Literature addressing this issue.

Line 115: Did they also have to do some sensory exercises? Please specify.

Tabe 1: Please check that it is viewed in one page.

Figure 5: It is not possible to read the legend.

Line 314: Would you expect that this difference is higher in the spring e.g., through flowering fields?

Line 317-319: Avoid redundancies. I have the feeling that you have mentioned this information already.

Line 348-350: Would you expect that the results are similar if more men were included? Did you also consider the "hormons" of women in your study? I may think that this influences also the results.

Line 369-375: The format of font size is not the same as before. Please check.

Line 379: "field, while depression 379 and anxiety rated with the POMS are less intense after staying on the field, compared to 380 the stay in the forest." Why is this? I think that you could discuss this aspect. more.

Author Response

Dear reviewer,

thank you very much for your comments. Please find our point by point response in the attachement.

Sincerely,

Katja Oomen-Welke

Reviewer 2 Report

This is an interesting article that comparing the reactions of two natural environments.

But some parts must be improved with more explanations and discussions.

1.     The explanation of the reason why HSPs were targeted seems inadequate and abrupt. Wasn't it necessary to find out what happens to the general population before selecting HSPs? On the other hand, are the results of this study specific to HSPs? Was the calming effect of the forest particularly effective as the authors predicted? How would the authors be predicted for the general population?

2.     More explanation is needed as to why the cultivated field was chosen for comparison with the forest.

3.     Additional explanation of the coloring of the graph is needed.

4.     In Figure 5, the right side of the text is cut off and the content is not clear. 5.

5.     It is difficult to understand from the Figure 5 what is the p=0.02. Is it the result of comparing the results after the forest intervention with those after the field intervention? Or is it the result of comparing the difference before and after the intervention? Similarly, what does the p-value in Table 2-4 compare? Is it a comparison of the amount of change before and after the intervention (T2-T1)?

6.     The mention of T1 and T2 appears for the first time in line250, and I couldn’t understand what it was at that time. In line271, T1 (before intervention) and T2 (after intervention) appear, but should be explained in line250.

7.     Was the difference in temperature between the forest and the field acceptable? The proximity of the distance is not really relevant; forest environments are generally cooler. If so, is there possibility that the heat in the fields caused discomfort and the difference was greater in the summer? If so, I think you should present the temperature and humidity data for discussion.

8.     What are possible reasons why cortisol did not make a difference?

9.     When the authors randomly assigned people to groups of up to 7 people, were there any considerations for age stratification or gender? I couldn’t figure it out how the group of 7 people each was combined into groups of 20 and 19 people, and what was the process?

10.  I believe it would have to be approved by an ethics committee, but there is no mention of the details.

Author Response

(The authors gave the same response as above.)

Round 2

Reviewer 2 Report

Most of the points have been corrected. However, the following problems have yet to be resolved.

Response 7: The difference in temperature between the two intervention sites was 0-2°C. Temperature was not consistently cooler in the forest, probably due to the local micro-climate. Relative humidity was higher in the forest, with differences of 5 to 20%. This can be attributed to usual forest climate. We regarded these differences as acceptable.

There is no basis for the author to determine that climate differences are acceptable. In the case of a study conducted in a natural environment, the authors should provide with information on the weather, temperature, and humidity at the time of the study.

Author Response

Dear reviewer,

more information about temperature and humidity was added in the discussion. Systematic measurements on each intervention day were not performed, which we mentioned as a limitation.

On behalf of all the authors,

Yours sincerely